

**Fossil-Dominated SOA Formation in Coastal China: Size-Divergent**
**Pathways of Aqueous Fenton Reactions versus Gas-phase VOC**
**Autoxidation**
Jia-Yuan Wang, Meng-Xue Tang, Shan Lu, Ke-Jin Tang, Xing Peng, Ling-Yan He, Xiao-Feng
Huang
Key Laboratory for Urban Habitat Environmental Science and Technology, School of Environment and
Energy, Peking University Shenzhen Graduate School, Shenzhen, 518055, China.
Corresponding author: Meng-Xue Tang (tangmx@pku.edu.cn)



**Abstract:** Elucidating size-dependent formation mechanisms of secondary organic aerosols (SOA)
remains a critical research gap in atmospheric chemistry. Here, we analyzed water-soluble compounds
in size-segregated aerosol samples (0.056–18 μm) collected at a coastal site in southern China.
Rradiocarbon ($^{14}C$) isotope analysis reveals that fossil sources dominate SOA in both fine (95.8%) and
coarse (80.4%) modes, while the small amount of biogenic SOA mostly existed in the coarse mode
(74.1%). Fine-mode oxygenated organic carbon (OOC) correlates strongly with polar carbonyl
compounds (e.g., glyoxal, methylglyoxal, acetone, and MVK+MACR), while coarse-mode OOC
exhibits better correlations with nonpolar aromatic hydrocarbons (e.g., toluene, C8 aromatic, C9 aromatic,
styrene) and biogenic VOCs (e.g., monoterpenes, isoprene), indicating that the sources of fine- and
coarse-mode OOC are different. Multivariate analyses incorporating inorganic ions, pH, water-soluble
iron ions, aerosol liquid water content, and $O_3$ revealed divergent size-dependent mechanisms,
emphasizing the significant role of aqueous-phase reactions in fine-mode OOC formation, particularly
the key contribution of water-soluble Fe ions ($r^2 = 0.74$), while coarse-mode OOC exhibited a notable
correlation with $O_3$ ($r^2 = 0.63$). Combining the information on VOCs precursors and key components,
our study elucidates that aqueous-phase reactions play a key role in fine-mode OOC, especially the
Fenton reaction, while gas-phase VOC autoxidation plays an important role in the coarse-mode OOC
generation. By examining OOC formation across a wide range of particle sizes, our study highlights the
critical need for mode-specific treatment of SOA generation in atmospheric chemical transport modeling.
**Key words:** Secondary organic aerosol (SOA), Fine mode, Coarse mode, Aqueous-phase reactions, Gas-
phase autoxidation



**Graphical abstract:**

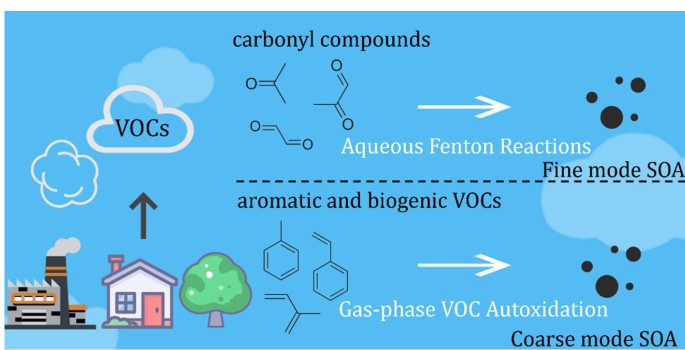






**1.Introduction**

In urban areas, organic aerosol (OA) constitutes 30-70% of submicron particle mass( Zhang et al.,

2017b) and has significant impacts on human health, radiation balance, and air quality. OA can originate
from direct emissions, known as primary organic aerosol (POA), or be formed in the atmosphere through
the oxidation of semi-volatile and volatile organic compounds (VOCs) followed by nucleation or
condensation of oxidation products onto the preexisting particles., resulting in secondary organic aerosol
(SOA) (Peng et al., 2021). Globally, SOA is estimated to contribute up to 93% of the total OA budget
(Hallquist et al., 2009). However, our understanding of the SOA formation mechanisms is still limited,
the complexities of SOA formation are not only due to the presence of large amounts of biogenic and
anthropogenic VOC precursors, but also because each VOC can undergo a number of atmospheric
degradation processes (e.g., gas-phase radical-mediated oxidation, heterogeneous oxidation, and
oligomerization) to produce various condensable oxidized organics (COO) with distinct functionality,
reactivity, and volatility(Gu et al., 2023; Xu et al., 2017; Yu et al., 2016).

Secondary organic aerosol (SOA) can be formed from the atmospheric oxidation of volatile organic

compounds (VOCs) or originate from various processes such as heterogeneous reactions, photochemistry,
and aqueous-phase oxidation (Dominutti et al., 2022). Field studies on SOA formation mostly focused
on fine particles ($PM_1$) partly because of instrument limitations (Xu et al., 2017; Yao et al., 2022a), recent
mass spectrometry-based studies have shown that photochemical oxidation has been suggested to be the
major pathway of SOA formation, photochemical oxidation of VOCs is generally initiated by reactions
with radicals (e.g., OH, $NO_3$) or oxidants (e.g., $O_3$), producing a variety of condensable oxidized organics
(COO) types, which subsequently engage in gas-to-particle conversion to contribute to SOA
formation(Xu et al., 2017; Zhan et al., 2021).However, aqueous-phase formation of SOA has also been



considered an important pathway, SOA can form in the aqueous phase on wet aerosols, clouds, and fogs
through further chemical processes involving water-soluble organic compounds or the organic products
of gas-phase photochemistry (Ervens et al., 2011; Gu et al., 2023; Mei et al., 2025).
While that formed on coarse particles was mostly neglected, dust (both natural and
anthropogenically emitted dust) is constantly present in the atmosphere and is one of the largest
contributors to aerosol mass in the troposphere(Wu et al., 2024; Xu et al., 2024), exerting a significant
impact on global climate by modulating radiative balance. Dust particles mainly consist of
aluminosilicate, sea salt, $SiO_2$, $CaCO_3$, and coated with secondary organic and inorganic aerosol
components under an ambient environment(Li & Shao, 2009; Yang et al., 2024), dust particles act as
reactants or catalysts, enhancing atmospheric heterogeneous reactions and photochemical processes(Pan
et al., 2023; Wang et al., 2020b).Heterogeneous reactions and photochemical reactions on mineral dust
may play an important role in coarse modal SOA generation (George et al., 2015; He et al., 2022a; Wang
et al., 2020b).This also suggests that different formation mechanisms may govern fine-mode and coarse-
mode secondary organic aerosols.
In this study, we collected a broad range of size-segregated samples (0.056-18 μm) from Shenzhen,
a coastal city in southern China, to obtain comprehensive particle size information. We utilized the offline
ACSM-PMF method to characterize SOA in these samples and combined it with [14]C analysis to gain a
deeper understanding of SOA from fossil fuel and biogenic sources (He et al., 2022a; Huang et al., 2020).
This study explores the mechanisms of SOA formation across both fine-mode and coarse-mode,
enhancing our understanding of the diverse generation mechanisms of SOA across various particle size
distributions.



## 2.Material and methods

### 2.1 Sampling site and sample collection

The sampling site, Atmospheric Observation Supersite of Shenzhen (AOSS, 22.60 ∘N, 113.98 ∘E), is located at an urban site in the southeast of the PRD region. There are no significant local pollution sources in the vicinity. The sampling period encompassed both the peak of particulate matter pollution and the most severe photochemical pollution in Shenzhen for the year. Atmospheric volatile organic compounds (VOCs) levels at this site are typically influenced by continental air masses, marine air masses, and local biogenic emissions(Li et al., 2024a). A ten-stage micro-orifice uniform deposit impactor (MOUDI, model 110, MSP Co., USA) with aerodynamic diameter cut-points of 0.056, 0.1, 0.18, 0.32, 0.56, 1.0, 1.8, 3.2, 5.6, 10, and 18 μm was used to collect size-segregated aerosol samples on Teflon filters from 21 October 2022 to 3 February 2023.In this study, we found that 1-1.8 μm particles showed more coarse mode properties, so we took 1 μm as the division boundary, so we use 1 μm as the as the boundary between fine particles and coarse particles, in this study. The sampling flow rate was 30 L/min. The average ambient temperature during the sampling period was 20.0 °C, and the dominant wind direction was northeasterly. In total, one hundred and sixty samples were collected with a sampling cycle of 72 hours.

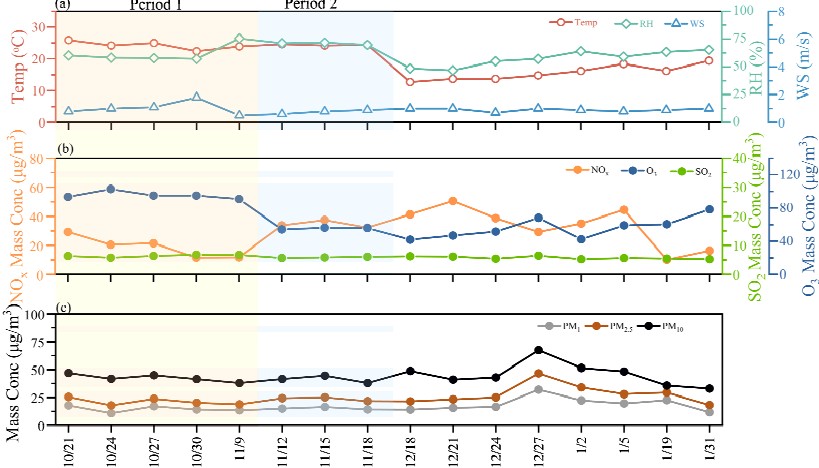

**Figure 1.** Time series of relative humidity (RH), Temperature (Temp) and wind speed (WS) (a), $O_3$, $SO_2$and $NO_x$ (b), $PM_1$, $PM_{2.5}$ and $PM_{10}$(c). The time series were categorized to be two typical periods based on total $O_3$ mass concentrations: the high-$O_3$ period (Period 1), and the low-$O_3$ period (Period 2)



**2.2. Chemical analysis**
The mass of the size-segregated aerosol samples was obtained from the difference in mass of the
Teflon filter before and after sampling in a cleanroom at conditions of 22.1 ℃ and 49.2% relative
humidity. Then, each filter was extracted with 20 mL of ultrapure water in an ultrasonic bath with ice for
30 min and then filtered with a 0.22 μm Teflon filter for further analysis. A portion of the water extract
was analyzed for water-soluble metal elements using an inductively coupled plasma mass spectrometer
(ICP-MS, Bruker auroraM90, Germany). Inorganic ions ($Cl^-$, $SO_4^{2-}$, $NO_3^-$, $NH_4^+$, $Na^+$, $K^+$, $Mg^{2+}$, $Ca^{2+}$)
were measured using an ion chromatography system (ICS-6000, Dionex, USA). A portion was analyzed
for water-soluble organic matter (WSOM) and the corresponding mass spectra using a Nebulizer-ACSM
system (ToF-ACSM-X, Aerodyne Research, Inc., USA) and a portion was analyzed for water-soluble
organic carbon (WSOC) with a total organic carbon analyzer (N/C 3100, Analytik Jena AG, Germany)
to quantify organic oxygen (WSOO, WSOM), the major ion fragments (m/z 44, m/z 57, m/z 65, m/z 93)
and elements(C, H, O, and N) measured by ToF-ACSM-X. Equal amounts of the water extract from the
same MOUDI stages were combined and concentrated for $^{14}C$ analysis based on accelerator mass
spectrometry. More details of the Nebulizer-ACSM and radiocarbon can be found in(He et al., 2022a;
Huang et al., 2020).
**2.3 Other measurements**
A meteorological monitoring instrument (WXT536, Vaisala, Finland) was used to measure the
meteorological variables, including atmospheric temperature (Temp), relative humidity (RH), wind
direction (WD), and wind speed (WS). Criteria air pollutants were monitored using the following
instruments: a 5030i $PM_{2.5}$ and 5030i $PM_{10}$ for particulate matter, a 43i $SO_2$ analyzer, a 42i NOx analyzer,
a 49i $O_3$ analyzer, and a 48i CO analyzer (Thermo Scientific, USA). Additionally, PTR-ToF-MS (6000X2,



Ionicon Analytik GmbH, Austria) with $H_3O^+$ ionization mode was used for online measurements of
volatile organic compounds at the same site during the campaign. Further details regarding the PTR-ToF-
MS are available in (He et al., 2022b; Li et al., 2024b).
**2.4 Data analysis**
The inorganic ion components of size-segregated aerosol samples ($Cl^-$, $SO_4^{2-}$, $NO_3^-$, $Na^+$, $NH_4^+$, $K^+$, $Mg^{2+}$,
$Ca^{2+}$), along with relative humidity (RH) and temperature were input into ISORROPIA II model to
calculate the aerosol liquid water content (ALWC) and aerosol pH ($pH_{aerosol}$), the thermodynamic
equilibrium model ISORROPIA II was used to estimate the size-resolved ALWC and $pH_{aerosol}$ in this
study owing to its accuracy, reliability, and high computational efficiency(Duan et al., 2020; Tan et al.,
2017; Xu et al., 2024). The Pearson correlation method was applied using SPSS Statistics software for
correlation analysis. Quantitative source apportionment of water-soluble organic carbon (WSOC) was
conducted with the U.S. EPA PMF v5.0 software. Data matrices and error matrix of WSOC, WSOO,
$CO_2^+$, $C_4H_9^+$, and nss-$K^+$ for a total of 160 samples (16 sets × 10 stages) were input into the PMF model,
the three-factor (the more oxidized oxygenated organic carbon (MO-OOC), the less oxidized OOC (LO-
OOC), and biomass-burning organic carbon (BBOC) determined to be the most reasonable solution
(Figure S1).More details of the source apportionment of WSOC by PMF modeling are provided in the
supporting information.
**3 Results and discussion**
**3.1 average size distributions of the aerosol components**
Figure 2a shows the average size distributions of the aerosol components, coarse modes exhibit higher
mass concentrations, accounting for 66.7% of the total mass. These coarse modes contain more water-



insoluble components, it contains a variety of metal oxides (i.e., $TiO_2$ and $Fe_2O_3$) (Adebiyi et al.,
2023).Unlike the coarse mode, the fine mode has a higher proportion of water-soluble components. As
is shown in Figure 2b, the main water-soluble inorganic ions in the fine mode differ from those in the
coarse mode, sulfate ($SO_4^{2-}$) and ammonium ($NH_4^+$) are the most abundant compounds in the fine mode,
constituting 17.0% and 7.4% of the total mass of fine particles, respectively. In contrast, nitrate ($NO_3^-$)
and calcium ($Ca^{2+}$) are the predominant inorganic ions in the coarse mode, comprising 5.6% and 1.5% of
the total mass of coarse particles, respectively. Although the compositions of fine- and coarse-mode
water-soluble inorganic ions differ significantly, water-soluble organic matter (WSOM) is the most
abundant water-soluble component in both modes. WSOM constitutes 55.9% of the total water-soluble
mass in fine particles and 40.9% in coarse particles, underscoring its critical role in both size modes.

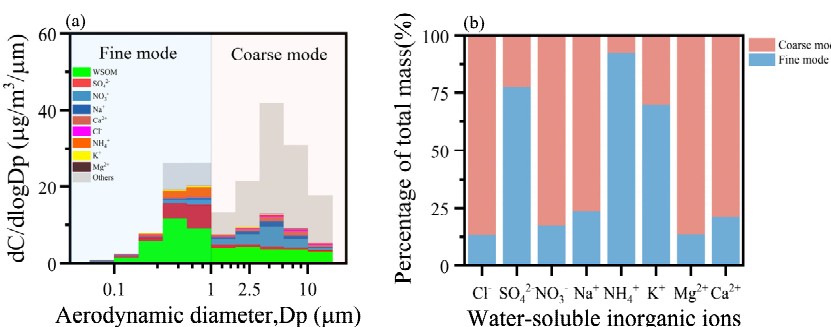


**Figure 2.** The average size distributions of aerosol components at this site (a), the percentage of the

fine mode and coarse mode of water-soluble inorganic ions (b).

**3.2 Possible sources of fine and coarse mode SOA**
In this study, PMF is used to extract OA components. Figure 3a shows the contributions of the
Oxygenated organic carbon (OOC; sum of MO-OOC and LO-OOC) and BBOC in all size bins. The
results indicate that BBOC was mainly disturbed in the fine mode accounting for 91.1% of the total




BBOC. OOC dominated in both the fine (64.4%) and coarse mode (88.4%), and previous studies found
that the fine mode SOA can be estimated by WSOC after removing the contribution of biomass
burning(He et al., 2022b; Huang et al., 2020), in this study , OOC is equivalent to SOA. This highlights
the critical role of SOA in both the fine and coarse mode.
Figure 3b shows the size distributions of fossil fuel OOC and biogenic OOC in all size bins, which were
calculated by combining the results from the PMF factor contributions and the $^{14}$C isotope analysis, and
the calculations were performed as in our previous study with equations (1)-(3) (He et al., 2022a;
Huangetal.,2020) :

$$\text{biogenic carbon} = \text{WSOC} * f_{modern} \tag{1}$$

$$\text{biogenic OOC} = \text{biogenic carbon} - \text{BBOC} \tag{2}$$

$$\text{fossil fuel OOC} = \text{LO-OOC} + \text{MO} - \text{OOC} - \text{biogenic OOC} \tag{3}$$

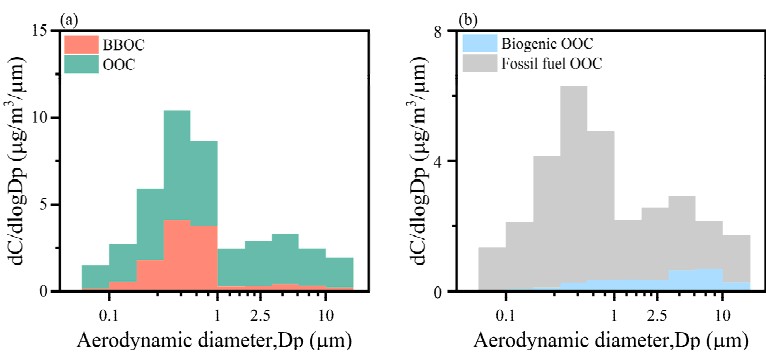

**Figure 3.** Average size distributions of the WSOC compositions, OOC and BBOC (a), fossil fuel OOC and biogenic OOC(b).

After removing the contributions from BBOC, the results clearly indicate that fossil fuel organic
carbon (OOC) dominates in both fine (95.8%) and coarse (80.4%) modes, reflecting the significant role





of anthropogenic sources in SOA generation. Regarding particle size distribution, fossil fuel OOC is
predominantly found in the fine mode (66.0%), while biogenic OOC is mainly present in the coarse mode
(74.1%). This distribution indicates the differing reaction pathways for SOA in fine and coarse modes.
We further explore the relationship between fine- and coarse-mode OOC and gaseous precursors,
more details of the gaseous precursors are provided in the supporting information (Table S1). We
observed a high correlation between fine-mode OOC and polar carbonyl compounds, such as glyoxal,
methylglyoxal, acetone, and MVK+MACR (Table 1). Carbonyl compounds are first- and/or second-
generation gas-phase oxidation products of both anthropogenic (e.g., aromatics, acetylene) and biogenic
(e.g., isoprene) sources (Ervens et al., 2011), this also suggests a complex source profile for fine mode
SOA. Additionally, carbonyl compounds have strong water solubility and can be absorbed into clouds
and fog to react with ·OH to form oligomers, which promote the formation of SOA(Wang et al.,
2022).However, unlike the fine mode, we found that coarse-mode OOC is uniquely correlated with
nonpolar aromatic hydrocarbons (e.g., toluene, C8 aromatic, C9 aromatic, styrene) and biogenic VOCs
(e.g., monoterpenes, isoprene) (Table 1). This revealed different gaseous precursors for fine- and coarse-
mode SOA, and reflected the different SOA generation mechanisms that may exist.
**Table 1.** The correlation coefficients between OOC and typical VOCs in the campaign. * indicates a
significance level of 95% ($p < 0.05$).

| | Monoterpenes | Isoprene | MVK+MACR | Toluene | C8 aromatic | C9 aromatic | Styrene | Glyoxal | Methylglyoxal | Acetone |
|---|---|---|---|---|---|---|---|---|---|---|
| Fine mode OOC | 0.20 | 0.47 | 0.70* | 0.38 | 0.39 | 0.36 | 0.46 | 0.70* | 0.73* | 0.62* |
| Coarse mode OOC | -0.75* | -0.56 | -0.34 | -0.60* | -0.65* | -0.66* | -0.74* | -0.39 | -0.39 | -0.52 |

**3.3 Possible formation mechanisms for fine mode SOA**
The previous results reveal a distinct origin for fine and coarse mode OOC, suggesting different
SOA generation mechanisms. Therefore, additional field measurements are necessary to further



understand the mechanisms and key factors affecting SOA formation.
Building on the findings from the previous section that fine-mode OOC are primarily derived from
carbonyl compounds, it is noteworthy that carbonyl compounds are highly reactive and exhibit
significant water solubility(Liu et al., 2022; Wang et al., 2022; Xu et al., 2022). These properties enable
them to contribute significantly to SOA formation through aqueous-phase reactions, particularly for
dicarbonyls such as glyoxal (Gly, CHOCHO) and methylglyoxal (Mgly, $CH_3COCHO$), which have been
identified as key SOA precursors (Liu et al., 2022; Tan et al., 2017) . Previous studies have identified
characteristic fragment ions of glyoxal and methylglyoxal (e.g., $C_2O_2^+$ and $CH_2O_2^+$), which play a crucial
role in the formation of low-volatility SOA during cloud processing and are strongly correlated with
aqueous oxygenated organic aerosol (aq-OOA) (Duan et al., 2020; Sun et al., 2016). As shown in Figure
4a, these fragment ions are predominantly distributed in fine particles, indicating the significance of
aqueous-phase processing in the fine mode. Further evidence of aqueous-phase reactions is provided by
the behavior of MVK and MACR. While the direct aqueous-phase reaction of MVK and MACR with
ozone is less competitive compared to the faster OH-initiated reactions (Chen et al., 2008) , aerosol liquid
water content (ALWC) serves as a key metric for characterizing aqueous-phase SOA formation due to
its positive correlation with these processes, especially under conditions of high relative humidity and
elevated $NO_x$ levels (Kuang et al., 2020b; McNeill, 2015; Zhan et al., 2021). In this study, we observed
a strong positive correlation between fine-mode OOC and ALWC (Figure 4b), suggesting that fine-
mode SOA is predominantly generated through aqueous-phase processes.
However, in contrast to the coarse mode, in contrast to the coarse modes, the fine modes are not
abundant in ALWC (Figure S4). Despite this, we observed a significant relationship between OOC and
ALWC exclusively in the fine mode. To further investigate the behavior of carbonyl compounds under



the unique conditions of the fine mode, we analyzed the correlations of fine- and coarse-mode OOC with
key factors, revealing notable differences between the two modes. First, the fine mode is characterized
by higher concentrations of inorganic ions, such as sulfate and nitrate, which may play a critical role in
SOA formation. Specifically, sulfate demonstrated a stronger positive influence on fine-mode OOC
formation compared to nitrate, as evidenced by their respective correlation coefficients ($R^2 = 0.85$ for
sulfate, Figure 4c; $R^2 = 0.47$ for nitrate, Figure 4d). This discrepancy may arise from the fact that sulfate
($SO_4^{2-}$) is primarily produced through aqueous-phase reactions, whereas nitrate ($NO_3^-$) is predominantly
generated via gas-phase reactions (Zhan et al., 2021) . Additionally, the fine mode exhibits acidic
conditions ($pH_{aerosol} = 0.4$–$4.3$), and we observed distinct correlations between fine-mode OOC and
$pH_{aerosol}$ (Figure 4e). This suggests that the lower pH in the fine mode favors the formation of fine-mode
OOC.

A few studies have emphasized the significant role of metal ions in SOA formation, particularly

under low pH conditions. To further investigate this, we examined the correlation between water-soluble
metal ions and fine-mode OOC (Table S5). Our analysis revealed that fine-mode OOC exhibits a strong
correlation with water-soluble Fe ions ($r = 0.82$, $p < 0.05$), and a positive relationship was observed
between the concentration of iron ions and fine-mode OOC (Figure 5f). Additionally, water-soluble iron
ions were found to be highly concentrated in the fine mode (Figure S4), with their concentration (18.87
$ng/m^3$ ) significantly exceeding that of other metal ions. Recent studies have highlighted the role of water-
soluble Fe ions in Fenton chemistry, where they cycle between $Fe^{2+}$ and $Fe^{3+}$. This process, particularly
through Fenton reactions involving peroxides, may substantially enhance SOA formation by supplying
particle-phase oxidants(Qin et al., 2022; Ye et al., 2021). Specifically, Fenton reactions within aqueous
particles can generate hydroxyl radicals (OH), which oxidize organic compounds such as carbonyls,



especially under lower pH conditions (Kuang et al., 2020a).

Therefore, in this study, we propose that aqueous-phase reactions play a dominant role in the formation

of fine-mode SOA. The lower pH and elevated concentrations of water-soluble Fe ions in the fine mode
create favorable conditions for SOA formation from carbonyl compounds, primarily through Fenton
reactions.

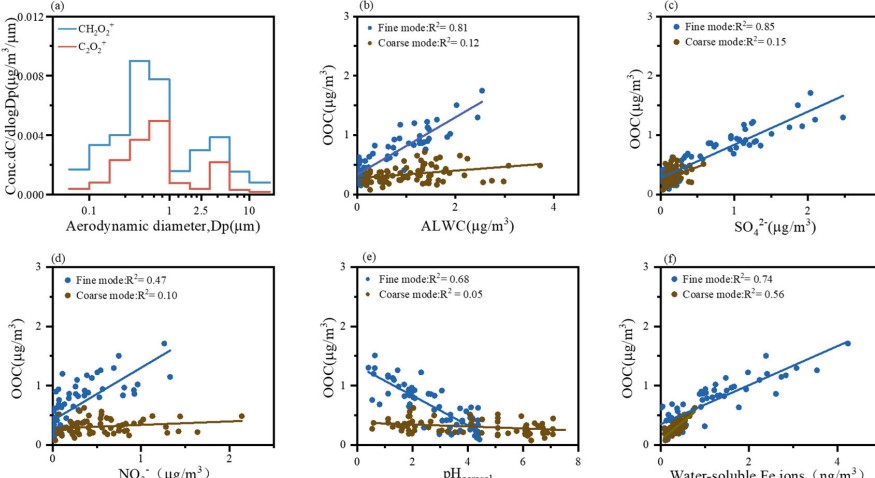


**Figure 4.** Average size distributions of $CH_2O_2^+$, and $C_2O_2^+$ (a), and relationship between OOC and

ALWC (b) , $SO_4^{2-}$(c), $NO_3^-$(d), $pH_{aerosol}$(e), water-soluble iron ions(f)

### 3.4 Possible formation mechanisms for coarse mode SOA

Most studies have focused on the heterogeneous uptake of inorganic trace gases on dust particles, and

few studies have attempted to investigate the uptake of VOCs on mineral dust particles. To date, the
impact of authentic dust particles on SOA growth has been poorly studied, the mechanistic role of the
mineral dust in SOA growth is uncertain under typical polluted urban environments(Yu et al., 2016; Yu
and Jang, 2019).

A distinct phenomenon observed in our experiments is that biogenic-OOC are predominantly

distributed in the coarse mode, which may be attributed to unique SOA formation pathways in this mode.



Biogenic-OOC is primarily generated through the oxidation of biogenic volatile organic compounds
(BVOCs) by atmospheric oxidants such as hydroxyl radicals (OH), ozone ($O_3$), and nitrate radicals ($NO_3$)
(Gagan et al., 2023). BVOCs emitted by terrestrial vegetation, including isoprene and monoterpenes,
significantly contribute to the total SOA budget. As shown in Table 1, coarse-mode OOC exhibits a strong
correlation with monoterpenes ($r = -0.75$, $p < 0.05$) but a weaker correlation with isoprene ($r = -0.56$, $p$
$< 0.05$), suggesting that monoterpenes play a more prominent role in biogenic-OOC formation.

Regarding the mechanisms of SOA generation from monoterpenes and isoprene, isoprene primarily

reacts with OH radicals to form SOA, whereas monoterpenes, in addition to reacting with OH radicals,
also undergo significant SOA formation through reactions with $O_3$ (McFiggans et al., 2019; Xu et al.,
2015). Previous studies have demonstrated that monoterpene-derived SOA is more oxidized in the
presence of nitrate-containing seed aerosols compared to ammonium sulfate seed aerosols (Huang et al.,
2016; Watne et al., 2017). The higher nitrate concentrations in the coarse mode further favor the $O_3$
oxidation pathway for monoterpenes. Our sampling site, located in the Pearl River Delta (PRD) region,
is one of the most rapidly urbanized areas with high anthropogenic emissions (Ma et al., 2024). The
sampling period coincided with elevated $O_3$ pollution levels. Coarse-mode particles, characterized by
higher pH compared to fine-mode particles, create conditions conducive to photosensitive reactions and
$O_3$ oxidation pathways    (Yu and Jang, 2019). Further analysis reveals a strong correlation between
coarse-mode OOC and $O_3$ (Figure 5a). Additionally, Figure 5b demonstrates that coarse-mode OOC
concentrations are significantly higher during high-$O_3$ periods compared to low-$O_3$ periods, with a
distinct peak observed during high-$O_3$ episodes. Notably, no significant increase in the concentrations of
other inorganic ions was observed during these high-$O_3$ periods (Figure S6). These findings collectively
underscore the critical role of $O_3$ in the formation of coarse-mode SOA.





However, the reaction pathways involved in coarse-mode secondary organic aerosol (SOA)
formation remain poorly understood. The $^{14}C$ isotope analysis results indicate that fossil fuel-derived
oxygenated organic compounds (OOC) are the primary source of coarse-mode OOC. Additionally,
coarse-mode OOC exhibits a stronger correlation with aromatic volatile organic compounds (VOCs),
particularly styrene (Table 1). However, since nonpolar aromatic hydrocarbons do not directly react with
$O_3$ to form SOA, further investigation is needed to elucidate the role of $O_3$ in coarse-mode SOA formation.
Recent studies have highlighted the rapid gas-phase autoxidation of endocyclic alkenes initiated by
ozonolysis, which yields highly oxygenated organic molecules (HOMs), particularly from monoterpenes
and aromatic compounds (Rissanen, 2021). Chemistry transport models have demonstrated that
ozonolysis of monoterpenes accounts for 79% of HOMs production (Shi et al., 2021). Additionally,
photochemical oxidation of substituted aromatic compounds has been shown to form HOMs through
rapid intramolecular autoxidation reactions, a process analogous to the oxidation of monoterpenes. $O_3$
can facilitate these reactions during the photo-oxidation of aromatics (Molteni et al., 2018; Suh et al.,
2003; Wang et al., 2020a) , which partially explains the stronger correlation between $O_3$ and coarse-mode
OOC. This conclusion is further supported by the higher oxidation state (O/C ratio) observed in the coarse
mode compared to the fine mode (Figure 5c).
Moreover, recent studies have identified carboxylic acids as products of these reactions (Zhang et
al., 2017a). The slope of coarse-mode OOC in the Van Krevelen (VK) plot is close to -0.5 (Figure 5d),
indicating the large formation of carboxylic acids with fragmentation through the replacement of
hydrogen atoms. The coarse mode is characterized by higher ALWC, higher pH, and favorable
partitioning of reaction products into the particulate phase. Based on these findings, we propose that gas-
phase autoxidation plays a significant role in the formation of coarse-mode SOA.



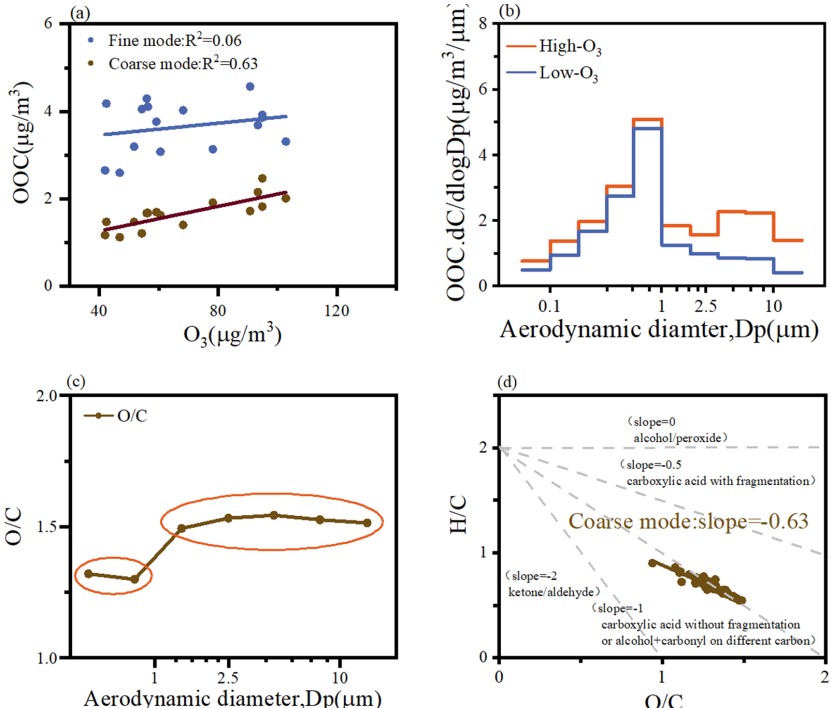

**Figure 5.** Relationship between OOC and $O_3$ (a), average size distributions of OOC (b), and organic

O/C(c), Van Krevelen diagram of H / C vs. O / C(d).

**4 Summary and implications**

This study collected 16 sets of size-segregated aerosol samples (0.056–18 μm) in Shenzhen, a

coastal city in the Pearl River Delta, from October 2022 to January 2023. The water-soluble components,

including typical inorganic ions, water-soluble organic compounds, and water-soluble metal ions, were

analyzed, and water-soluble organic matter (WSOM) emerged as the most abundant water-soluble

component in both modes, accounting for 55.9% and 40.9% of the total water-soluble mass in fine and

coarse particles, respectively. This highlights the critical role of WSOM in both size fractions.

Our findings indicate that WSOM in both fine and coarse modes exhibits secondary production. To



quantify secondary organic aerosol (SOA), which is represented by oxygenated organic carbon (OOC),
we applied Positive Matrix Factorization (PMF) modeling and utilized radiocarbon isotopes to
distinguish between fossil fuel-derived and biogenic organic carbon (OOC). Rradiocarbon ($^{14}$C) isotope
analysis reveals that fossil sources dominate SOA in both fine (95.8%) and coarse (80.4%) modes, while
the small amount of biogenic SOA mostly existed in the coarse mode (74.1%), we emphasize the
significant contribution of anthropogenic volatile organic compounds (VOCs) to SOA formation in
coastal atmospheres, where high relative humidity and enhanced atmospheric oxidation capacity also
play pivotal roles in SOA generation across both fine and coarse modes. Furthermore, we investigated
potential precursor sources for fine- and coarse-mode OOC, fine-mode oxygenated organic carbon (OOC)
correlates strongly with polar carbonyl compounds (e.g., glyoxal, methylglyoxal, acetone, and
MVK+MACR), while coarse-mode OOC exhibits better correlations with nonpolar aromatic
hydrocarbons (e.g., toluene, C8 aromatic, C9 aromatic, styrene) and biogenic VOCs (e.g., monoterpenes,
isoprene), indicating that the sources of fine- and coarse-mode OOC are different, indicating that the
sources of fine- and coarse-mode OOC are different, indicating distinct precursor sources for SOA in
different size modes.

Multivariate analyses incorporating inorganic ions, pH, water-soluble Fe ions, aerosol liquid water

content, and O$_3$ revealed divergent size-dependent mechanisms, emphasizing the significant role of
aqueous-phase reactions in fine-mode OOC formation, particularly the key contribution of water-soluble
iron ions (r$^2$ = 0.74), while coarse-mode OOC exhibited a notable correlation with O$_3$ (r$^2$ = 0.63).
Combining the information on VOCs precursors and key components, our study elucidates that aqueous-
phase reactions play a key role in fine-mode OOC, especially the Fenton reaction, while gas-phase
autoxidation plays an important role in the coarse-mode OOC generation. By examining OOC formation



across a wide range of particle sizes, this study provides novel insights into SOA formation mechanisms
and enhances our understanding of the formation pathways of SOA in both fine and coarse mode.
However, the specific mechanisms governing SOA generation in different particle size ranges remain
poorly understood. We strongly recommend further laboratory experiments to explore these mechanisms
in greater depth. Notably, our study underscores the significant role of anthropogenic VOCs in SOA
formation in coastal environments, where high relative humidity and atmospheric oxidation capacity are
critical drivers. Similar conditions are prevalent in marginal seas and estuaries near urban areas,
warranting further in-depth studies in these representative regions.



**Data availability.** Datasets are available by contacting the corresponding author, Meng-Xue Tang
(tangmx@pku.edu.cn)
**Supplement.** The supplement material related to this article is available online at:
**Author contributions.** WJ,TM and HX conceptualized the study. WJ, LS, and TJ executed the
experiments. WJ and TM carried out the statistical analysis. WJ prepared the first draft of the manuscript,
which was commented on and revised by TM and HX. All authors reviewed and approved the final
version for publication.
**Competing interests.** The authors declare that they have no conflict of interest.
**Financial support.** This work was supported by the National Key Research and Development Program
of China (2022YFC3701000, Task2) and the National Natural Science Foundation of China (42407145).





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
