# Peer review of "Fossil-Dominated SOA Formation in Coastal China: Size-Divergent"

_EGUsphere, 2025_

## Author Comment (AC1)

We are very grateful for the anonymous reviewer's positive assessments of the manuscript and insightful comments for further improvement. We have revised the manuscript by fully taking the reviewers' suggestions into account. Please find our point-to-point replies below in blue, and the specific changes in the revised manuscript and SI are highlighted here in red.

**Reviewer 1**

This manuscript presents analysis of size-segregated water-soluble aerosol samples from a coastal site in China. The main findings include that nearly all OOC was derived from fossil sources, fine-mode SOA was influenced by aqueous phase Fenton reactions and coarse mode SOA was derived from gas-phase oxidation. I have a few major comments that need to be addressed before the manuscript is ready for publication.

1. I have major concerns regarding the PMF analysis. This has implications for the overall conclusions as the authors have defined SOA as the sum of two of the PMF species. Specifically:

   Why were only 5 variables included, rather than including more species from the IC, ICP or ACSM?

   The would strengthen the separation of the factors. How was nss-K derived? Based on the high Cl and $NO_3$ in the coarse mode, it seems there is probably influence from sea salt.

   The PMF results are also concerning and more validation is necessary. I would expect at least some $CO_2^+$ and $C_4H_9^+$ to be attributed to biomass burning. It would be useful to also show the rest of the results from the PMF rather than just the EV plot.

**Response:**

Thank you very much for your thoughtful and detailed comments regarding the PMF analysis. We appreciate your suggestions, and we respond to each of your concerns below:

(1) Regarding the number of variables used in PMF.

Thank you very much for your valuable comments. We fully understand your concerns regarding the limited number of variables used in the PMF analysis. In this study, our primary aim was to investigate the formation mechanisms and size distribution characteristics of secondary organic aerosols (SOA). Therefore, we intentionally selected representative variables that are closely related to SOA formation, such as WSOC, WSOO, and key organic fragments from the ACSM, in order to enhance the interpretability of the SOA-related factors extracted by the PMF model. In our current study, we selected five variables for PMF analysis based on the methodology and experience from our group's previous research (e.g.,(He et al., 2022; Huang et al., 2020; Wei et al., 2024)), where robust and interpretable factor solutions were achieved using this approach.

While species from IC, ICP, or additional ACSM ions may indeed be important in broader aerosol characterization, our current focus was to minimize uncertainties related to multicollinearity and to concentrate specifically on resolving SOA-related sources. Including too many overlapping chemical tracers could increase model complexity and hinder the effective extraction of interpretable SOA factors. That said, we agree that incorporating a wider range of species could potentially help further resolve mixed sources and their contributions to SOA formation. We will consider this in future studies.

(2) Derivation of nss-K$^+$:

Thank you for your comments. In response to the reviewer's concern regarding the calculation of non-sea-salt potassium (nss-K$^+$), we have clarified this in the revised manuscript. Specifically, we have added the following sentence in the main text (Lines 132-135): "Considering the significant contribution of sea salt at the sampling site, non-sea-salt potassium (nss-K$^+$) was calculated to better represent biomass burning emissions. Nss-K$^+$ was calculated from measured K$^+$ assuming the mass ratio K$^+$/Na$^+$ of 0.036 as in seawater, following the approach in (Boreddy and Kawamura, 2015; Klopper et al., 2020)."

(3) Clarification of PMF results:

We agree that displaying the full factor profiles could enhance transparency and provide a more comprehensive view of the PMF outputs. We will include the complete mass spectral profiles of all resolved factors in the Supplementary Information in the revised manuscript. The added PMF results and evaluation for WSOC (line 44-50 in the revised SI) are as follows:

"Two to five factors were tested for modeling, and the three-factor output (base run, $Q_{true}/Q_{exp}$=1.01) was found to be the most reasonable solution to explain all identified factor profiles, as will be discussed later; additionally, the solutions with more than three factors did not produce any new meaningful results for WSOC. The scaled residuals exhibited a generally symmetrical distribution between -3 and +3 as well. Moreover, there was also a strong overall correlation between the total factor concentrations reconstructed by the PMF model and the total mass concentrations of the measured species (Figure S4)."

In addition, Figure S3 and Figure S4 was also added to SI to verify the PMF results.

[Figure]

Figure S3. The concentrations of species with standard deviations for the three-factor solution resolved in the PMF analysis.

[Figure]

Figure S4. Comparison between the measured total mass of species and the PMF-reconstructed total mass of sources for WSOC(a), WSOO(b), $CO_2^+$(c), $C_4H_9^+$(d), and nss-$K^+$(e).

We appreciate the reviewer's insightful comments and agree that further clarification regarding the BBOC factor is necessary. In our PMF analysis, non-sea-salt potassium (nss-$K^+$), a well-established tracer for biomass burning, was primarily attributed to Factor 1, which had an intermediate O/C ratio of 0.53. This is consistent with previous studies identifying aged biomass burning organic carbon (BBOC) with similar oxidation levels (Feng et al., 2023; Sun et al., 2018).

Although $CO_2^+$ and $C_4H_9^+$ were not predominantly associated with the BBOC factor, this can be reasonably explained. $CO_2^+$ is a common fragment in both highly oxidized SOA and aged BBOA. In our dataset, $CO_2^+$ was largely assigned to the MO-OOC factor, which exhibited a very high O/C ratio (1.85). This is consistent with $CO_2^+$ being a generic marker for highly oxidized compounds rather than a source-specific tracer (Dai et al., 2019; Xu et al., 2019).

As for $C_4H_9^+$, it was mainly allocated to the LO-OOC factor. While this fragment is often used as a tracer for hydrocarbon-like organic aerosol (HOA), previous studies have also noted its presence in less oxidized fractions of biomass burning emissions (Cao et al., 2018; Hu et al., 2017). Given the overlapping fragmentation patterns between fresh HOA and certain primary BBOA components, it is not unexpected that $C_4H_9^+$ was not uniquely associated with the BBOC factor in our solution. Thus, we believe the current attribution is reasonable within the resolution and scope of our PMF analysis.

We believe these clarifications and additional data presentation strengthen the reliability of our PMF interpretation and the conclusions drawn about SOA sources. All changes have been noted in the revised manuscript.

2. Should the aerosol size distribution units be $\mu g \cdot m^{-3}$?

**Response:**

Thanks for your suggestion. In our manuscript, we used dC/dlogDp ($\mu g \cdot m^{-3}$) to present size-resolved aerosol mass distributions, which reflects the differential concentration per logarithmic particle diameter interval. This is a standard approach when illustrating aerosol size distribution patterns. For figures or text where size resolution is not emphasized, we used $\mu g \cdot m^{-3}$ to represent the bulk mass concentration of aerosol components. We believe this dual usage is appropriate to clearly distinguish between size-resolved and total concentrations.

3. According to table 1, coarse mode OOC is negatively correlated with the aromatic VOCs. Similarly at line 255, the correlation between coarse mode biogenic OOC and monoterpenes and isoprene is discussed, but the correlations are negative. I'm not sure how this supports these VOCs as precursors for the coarse mode OOC.

**Response:**

Thank you very much for your insightful comments. We fully understand your concerns regarding the negative correlations observed between coarse-mode OOC and aromatic VOCs as well as biogenic VOCs (monoterpenes and isoprene). We appreciate the opportunity to clarify this important issue. Here, we provide detailed explanations addressing your concerns and reinforcing our conclusion that aromatic VOCs and biogenic VOCs are indeed significant precursors for coarse-mode SOA, despite the observed negative correlations.

(1) Size-specific correlations support distinct SOA precursor pathways

In our study, coarse-mode OOC shows statistically significant correlations (albeit negative) with aromatic VOCs (e.g., toluene, styrene) and biogenic VOCs (e.g., monoterpenes, isoprene). Notably, these correlations are uniquely observed for coarse-mode OOC and are not present for fine-mode OOC. Such size-specific relationships clearly suggest distinct precursors and formation mechanisms for fine- and coarse-mode SOA, which strongly supports our conclusion that aromatic VOCs and BVOCs serve as important precursors for coarse-mode OOC.

(2) Direct evidence from [14]C isotope analysis

Our [14]C analysis clearly distinguishes the origins of coarse-mode OOC, indicating a mixed contribution of fossil sources (approximately 80.4%) and biogenic sources (approximately 19.6%). The correlations observed align well with these results, where aromatic VOCs correspond predominantly to fossil-derived OOC and biogenic VOCs (monoterpenes and isoprene) correspond to biogenic-derived OOC. This strong agreement between source attribution and VOC correlations directly supports our conclusion that these VOCs are valid and significant precursors of coarse-mode SOA.

(3) Experimental evidence: Biogenic OOC predominantly occurs in the coarse mode

Our analysis explicitly demonstrates that biogenic OOC is primarily found in coarse-mode particles (74.1%), providing direct observational evidence that oxidation products derived from biogenic VOCs (monoterpenes and isoprene) preferentially accumulate in coarse-mode aerosols. This further substantiates the critical role of BVOCs as important coarse-mode SOA precursors.

To clarify these important points, we have added the following sentence in the main text (Lines 196-207): "In contrast, coarse-mode OOC exhibited significant negative correlations with nonpolar aromatic hydrocarbons (e.g., toluene, C8 aromatic, C9 aromatic, styrene) and biogenic VOCs (monoterpenes) (Table 1). Despite these negative correlations, several evidence support atmospheric relevance of these gaseous precursors to coarse-mode SOA. Firstly, the correlations with aromatic and biogenic VOCs were unique to coarse-mode OOC and not observed in the fine-mode OOC, clearly demonstrating distinct precursor pathways for coarse and fine-mode SOA. Secondly, $^{14}$C isotope analysis explicitly confirmed that coarse-mode OOC consisted of both fossil (approximately 80.4%) and biogenic (approximately 19.6%) components, directly aligning with the respective aromatic and biogenic VOC precursors identified here. Thirdly, biogenic OOC was found predominantly in coarse-mode particles (74.1%), providing direct observational evidence linking biogenic VOC oxidation products to coarse-mode aerosol formation."

We sincerely appreciate the reviewer's insightful comment, which allowed us to refine our manuscript and better convey our scientific conclusions.

4. Line 61: "Dust particles mainly consist of aluminosilicate, sea salt, SiO2, CaCO3, and coated with secondary organic and inorganic aerosol components under an ambient environment…" I would argue that sea salt particles are, by definition, not dust.

**Response:**

Thank you for your valuable comment. We have carefully reviewed the sentence in question and agree that using "dust" to refer to coarse particulates may lead to confusion, as sea salt and other particles are not technically classified as dust. Therefore, we have revised the sentence to better reflect the composition of coarse particles. The updated sentence (Lines 60-62) now reads: "Coarse particles mainly consist of aluminosilicate, $SiO_2$, $CaCO_3$, sea salt, and coated with secondary organic and inorganic aerosol components under an ambient environment."

5.  Figure 2: What is included in the "Other" category?

**Response:**

Thanks for your suggestion. Detailed explanations of the "Others" have been added to the caption of Figure 2 (lines 154-156) in the revised manuscript, as presented below: 'The "others" category was calculated by the mass concentration of particulate matter minus the total concentrations of water soluble species, and might include non-water soluble organic matter, elemental carbon, crustal material, etc.'

6.  Line 138: "These coarse modes contain more water insoluble components, it contains a variety of metal oxides (i.e., $TiO_2$ and $Fe_2O_3$)". Is this just a general comment, or were these measured? If so, how were these water-insoluble compounds measured?

**Response:**

Thanks for your comments. Following your suggestion, I have revised the relevant sentence (Lines 139-141) as follows: "The coarse mode contains significantly higher proportions of water-insoluble components, with measured concentrations reaching 20.63 $\mu g\ m^{-3}$, accounting for 75.6% of the total coarse-mode mass concentration."

7.  Line 163: Please include more details for these equations. For example, what is $f_{modern}$?

**Response:**

Thanks for your suggestion. The clarification of the equations has been provided in the revised manuscript, as presented below (Lines 175-180):"Here, $f_{modern}$ represents the modern carbon fraction, defined as the ratio of the $^{14}C/^{12}C$ content in a sample relative to that of a modern standard (NBS Oxalic Acid I from AD 1950), corrected for $\delta^{13}C$ isotopic fractionation and $^{14}C$ decay(Zhang et al., 2019). Biogenic carbon represents the portion of carbon derived from biogenic sources, biogenic OOC represents the oxygenated organic carbon originating from biogenic sources, fossil fuel OOC represents the oxygenated organic carbon derived from fossil fuel sources. "

The clarification of the equations (Lines 165-167) has been provided in detail in Supplementary Text S2, which includes the following description (Lines 65-75 in the revised SI): "The dried filters were used to make graphite samples using the graphitization line at the Guangzhou Institute of Geochemistry, CAS through the hydrogen and zinc reduction method(Xu et al., 2007), and then graphite samples were measured with a compact accelerator mass spectrometry (NEC, National Electrostatics Corporation, USA) at the Guangzhou Institute of Geochemistry, CAS. AMS calibration was performed using standards (Oxalic Acid Standards I and II) and blanks. The $\delta^{13}C$ value was obtained during AMS measurements and applied to correct the $^{14}C$ measurements for isotopic fractionation. The fraction modern ($f_{modern}$) )was determined by comparing the measured $^{14}C/^{12}C$ ratio in a sample with that in a modern standard (NBS Oxalic Acid I in AD 1950). All of the reported fm values were corrected for $\delta^{13}C$ fractionation and for $^{14}C$ decay over the time period between 1950 and the year of measurement and more technical details can be found in the literatures (Zhang et al., 2019; Zhu et al., 2015)."

8. Line 198: Are these ion fragments from the ACSM or the PTR-MS?

**Response:**

Thanks for your suggestion. Thank you for your question. The characteristic fragment ions of glyoxal and methylglyoxal (e.g., $C_2O_2^+$ and $CH_2O_2^+$) mentioned in Line 198 were identified in previous studies using aerosol mass spectrometry techniques, including both the aerosol mass spectrometer (AMS) and the aerosol chemical speciation monitor (ACSM). We have revised the sentence accordingly in Lines 220-224 to clarify this point,the revised text is as follows: "Previous studies have identified characteristic fragment ions of glyoxal and methylglyoxal (e.g., $C_2O_2^+$ and $CH_2O_2^+$), detected by aerosol mass spectrometer (AMS) and aerosol chemical speciation monitor (ACSM), which play a crucial role in the formation of low-volatility SOA during cloud processing and are strongly correlated with aqueous oxygenated organic aerosol (aq-OOA) (Duan et al., 2020; Sun et al., 2016)."

9. Line 13: Radiocarbon is misspelled

**Response:**

Thank you for pointing this out. We have corrected the spelling of "Radiocarbon" in Line 13.

10. SOA and VOC were already defined previously

**Response:**

Thank you for your comment. We acknowledge that SOA and VOC were already defined previously. The redundant definitions have been removed in the revised manuscript to improve clarity and avoid repetition.

11. "While that formed on coarse particles was mostly neglected…" this clause is not clear

**Response:**

Thank you for your valuable feedback. We have revised the sentence to clarify the point regarding the formation of SOA on coarse particles. The revised sentence (line 57) now reads: "While the formation of SOA on coarse particles was mostly neglected". This modification makes the sentence clearer and directly addresses the focus of our study on coarse-mode SOA formation. We hope this revision improves the clarity of the manuscript.

12. PRD is not defined

**Response:**

Thank you for pointing this out. We have added the full definition of PRD (Pearl River Delta) when it first appears in the manuscript to improve clarity for readers who may not be familiar with the abbreviation. The revised sentence(line 77-78) now reads: "The sampling site, Atmospheric

Observation Supersite of Shenzhen AOSS(22.60 °N, 113.98 °E), is located at an urban site in the southeast of the Pearl River Delta (PRD) region."

**Reference**

Boreddy, S. K. R. and Kawamura, K.: A 12-year observation of water-soluble ions in TSP aerosols
collected at a remote marine location in the western North Pacific: an outflow region of Asian dust,
Atmospheric Chem. Phys., 15, 6437–6453, 2015.

Cao, L.-M., Huang, X.-F., Li, Y.-Y., Hu, M., and He, L.-Y.: Volatility measurement of atmospheric
submicron aerosols in an urban atmosphere in southern China, Atmospheric Chem. Phys., 18, 1729–
1743, 2018.

Dai, Q., Schulze, B. C., Bi, X., Bui, A. A. T., Guo, F., Wallace, H. W., Sanchez, N. P., Flynn, J. H., Lefer,
B. L., Feng, Y., and Griffin, R. J.: Seasonal differences in formation processes of oxidized organic
aerosol near Houston, TX, Atmospheric Chem. Phys., 19, 9641–9661, 2019.

Duan, J., Huang, R.-J., Li, Y., Chen, Q., Zheng, Y., Chen, Y., Lin, C., Ni, H., Wang, M., Ovadnevaite, J.,
Ceburnis, D., Chen, C., Worsnop, D. R., Hoffmann, T., O'Dowd, C., and Cao, J.: Summertime and
wintertime atmospheric processes of secondary aerosol in Beijing, Atmospheric Chem. Phys., 20,
3793–3807, 2020.

Feng, T., Wang, Y., Hu, W., Zhu, M., Song, W., Chen, W., Sang, Y., Fang, Z., Deng, W., Fang, H., Yu, X.,
Wu, C., Yuan, B., Huang, S., Shao, M., Huang, X., He, L., Lee, Y. R., Huey, L. G., Canonaco, F.,
Prevot, A. S. H., and Wang, X.: Impact of aging on the sources, volatility, and viscosity of organic
aerosols in Chinese outflows, Atmospheric Chem. Phys., 23, 611–636, 2023.

He, D.-Y., Huang, X.-F., Wei, J., Wei, F.-H., Zhu, B., Cao, L.-M., and He, L.-Y.: Soil dust as a potential
bridge from biogenic volatile organic compounds to secondary organic aerosol in a rural
environment, Environ. Pollut., 298, 118840, 2022.

Hu, W., Hu, M., Hu, W.-W., Zheng, J., Chen, C., Wu, Y., and Guo, S.: Seasonal variations in high time-
resolved chemical compositions, sources, and evolution of atmospheric submicron aerosols in the
megacity Beijing, Atmospheric Chem. Phys., 17, 9979–10000, 2017.

Huang, X.-F., Dai, J., Zhu, Q., Yu, K., and Du, K.: Abundant Biogenic Oxygenated Organic Aerosol in
Atmospheric Coarse Particles: Plausible Sources and Atmospheric Implications, Environ. Sci.
Technol., 54, 1425–1430, 2020.

Klopper, D., Formenti, P., Namwoonde, A., Cazaunau, M., Chevaillier, S., Feron, A., Gaimoz, C., Hease,
P., Lahmidi, F., Mirande-Bret, C., Triquet, S., Zeng, Z., and Piketh, S. J.: Chemical composition and
source apportionment of atmospheric aerosols on the Namibian coast, Atmos Chem Phys, 2020.

Sun, Y., Du, W., Fu, P., Wang, Q., Li, J., Ge, X., Zhang, Q., Zhu, C., Ren, L., Xu, W., Zhao, J., Han, T.,
Worsnop, D. R., and Wang, Z.: Primary and secondary aerosols in Beijing in winter: sources,
variations andprocesses, Atmospheric Chem. Phys., 16, 8309–8329, 2016.

Sun, Y., Xu, W., Zhang, Q., Jiang, Q., Canonaco, F., Prévôt, A. S. H., Fu, P., Li, J., Jayne, J., Worsnop,
D. R., and Wang, Z.: Source apportionment of organic aerosol from 2-year highly time-resolved
measurements by an aerosol chemical speciation monitor in Beijing, China, Atmospheric Chem.
Phys., 18, 8469–8489, 2018.

Wei, F., Peng, X., Cao, L., Tang, M., Feng, N., Huang, X., and He, L.: Characterizing water solubility of
fresh and aged secondary organic aerosol in $PM_{2.5}$ with the stable carbon isotope technique,
Atmospheric Chem. Phys., 24, 8507–8518, 2024.

Xu, W., Xie, C., Karnezi, E., Zhang, Q., Wang, J., Pandis, S. N., Ge, X., Zhang, J., An, J., Wang, Q.,
Zhao, J., Du, W., Qiu, Y., Zhou, W., He, Y., Li, Y., Li, J., Fu, P., Wang, Z., Worsnop, D. R., and Sun,
Y.: Summertime aerosol volatility measurements in Beijing, China, Atmospheric Chem. Phys., 19,
10205–10216, 2019.

Xu, X., Trumbore, S. E., Zheng, S., Southon, J. R., McDuffee, K. E., Luttgen, M., and Liu, J. C.:
Modifying a sealed tube zinc reduction method for preparation of AMS graphite targets: Reducing
background and attaining high precision, Nucl. Instrum. Methods Phys. Res. Sect. B Beam Interact.
Mater. At., 259, 320–329, 2007.
Zhang, X., Li, J., Mo, Y., Shen, C., Ding, P., Wang, N., Zhu, S., Cheng, Z., He, J., Tian, Y., Gao, S., Zhou,
Q., Tian, C., Chen, Y., and Zhang, G.: Isolation and radiocarbon analysis of elemental carbon in
atmospheric aerosols using hydropyrolysis, Atmos. Environ., 198, 381–386, 2019.
Zhu, S., Ding, P., Wang, N., Shen, C., Jia, G., and Zhang, G.: The compact AMS facility at Guangzhou
Institute of Geochemistry, Chinese Academy of Sciences, Nucl. Instrum. Methods Phys. Res. Sect.
B Beam Interact. Mater. At., 361, 72–75, 2015.

---

## Author Comment (AC2)

We are very grateful for the anonymous reviewer's positive assessments of the manuscript and insightful comments for further improvement. We have revised the manuscript by fully taking the reviewers' suggestions into account. Please find our point-to-point replies below in blue, and the specific changes in the revised manuscript and SI are highlighted here in red.

**Reviewer 2**

This manuscript is an interesting effort to understand secondary organic aerosol formation in a coastal site in China. It could serve as a valuable guide for further complementary studies on the differences between fine-mode and coarse-mode oxygenated organic carbon origins. I recommend minor revisions for publication. Additionally, the text should undergo a careful review for grammar and fluency, with particular attention to punctuation and spaces.

1. Why not utilize diagnostic ratios to attribute and support specific sources such as $Mg^{2+}/Na^+$, $Cl^-/Na^+$ (related to marine influence), and $SO_4^{2-}/NO_3^-$ (which some authors use to differentiate between stationary and vehicular sources) among the various particle sizes? It can be complementary to PMF.

**Response:**

Thank you very much for your insightful comment. We fully understand your suggestion regarding the use of diagnostic ratios (e.g., $Mg^{2+}/Na^+$, $Cl^-/Na^+$, and $SO_4^{2-}/NO_3^-$) to support source identification across particle sizes. However, the primary objective of this study was to investigate the formation mechanisms and size distribution of secondary organic aerosols (SOA), with a particular focus on water-soluble organic carbon (WSOC). Therefore, the PMF analysis was conducted using variables closely related to SOA, such as WSOC, WSOO, and selected organic fragments from the ACSM, to ensure the interpretability of SOA-related factors.

This approach follows the methodology successfully applied in our previous work (e.g., He et al., 2022; Huang et al., 2020; Wei et al., 2024), where robust and meaningful SOA source apportionment was achieved using a limited number of representative variables. While diagnostic ratios and additional species may indeed help in identifying other aerosol sources such as sea salt or anthropogenic sulfate, including too many variables could increase model uncertainty and reduce the clarity of SOA-related factors. We appreciate your suggestion and will consider incorporating such diagnostic indicators in future studies that aim to provide more comprehensive source apportionment.

2. Please provide the robustness assessment of the Positive Matrix Factorization (PMF) results, including bootstrap mapping and displacement tests, and clarify how the three-factor solution was determined (in the manuscript).

**Response:**

We appreciate the reviewer's suggestion. In response, we have clarified the rationale for selecting the three-factor solution in the revised manuscript. The revised sentence(line 158-160) now reads: "The three-factor solution was considered the most reasonable based on the clarity of factor profiles and the residual distribution. Further details are provided in the Supplement."

We have provided a detailed robustness assessment of the PMF results in the revised Supplementary Information, including both bootstrap and displacement tests. As shown in the newly added Table S1 and described in Supplementary Text S1(Lines 50-53 in the revised SI), "All three factors were successfully mapped in 100% of the bootstrap (BS) runs, and no factor swaps were observed in the displacement (DISP) test. The absence of swaps indicates that the PMF results are sufficiently robust (Table S1)." These additions have been included to enhance the credibility and reproducibility of our factor identification.

**Table S1.** Diagnostic parameters of BS and DISP error estimates of three factors of source analytic results of PMF model

| diagnostics | Diagnostic parameters | 3 factors |
|---|---|---|
| BS diagnostics | % BS mapping | 100% |
| | % Unmapped | 0 |
| | Error Code | 0 |
| DISP diagnostics | Largest Decrease in Q | 0 |
| | %dQ | <0.1% |
| | Swaps by Factor | 0 |

Finally, in response to your suggestion that "the text should undergo a careful review for grammar and fluency, with particular attention to punctuation and spaces," we have thoroughly proofread and revised the manuscript to address these issues and enhance the overall clarity and language quality.

**Reference**

He, D.-Y., Huang, X.-F., Wei, J., Wei, F.-H., Zhu, B., Cao, L.-M., and He, L.-Y.: Soil dust as a potential bridge from biogenic volatile organic compounds to secondary organic aerosol in a rural environment, Environ. Pollut., 298, 118840, 2022.

Huang, X.-F., Dai, J., Zhu, Q., Yu, K., and Du, K.: Abundant Biogenic Oxygenated Organic Aerosol in Atmospheric Coarse Particles: Plausible Sources and Atmospheric Implications, Environ. Sci. Technol., 54, 1425–1430, 2020.

Wei, F., Peng, X., Cao, L., Tang, M., Feng, N., Huang, X., and He, L.: Characterizing water solubility of fresh and aged secondary organic aerosol in PM$_{2.5}$ with the stable carbon isotope technique, Atmospheric Chem. Phys., 24, 8507–8518, 2024.